Morphological and structural complexity analysis of low-resource English-Turkish language pair using neural machine translation models

Acı Mehmet
Vuran Sarı Nisa nvuran@mersin.edu.tr
http://orcid.org/0000-0002-0028-9890 İnan Acı Çiğdem
Department of Computer Engineering, Mersin University , Mersin , Turkey
Angiulli Giovanni
Electronic publication date: 2025 Aug 11
Publication date: 2025
Volume: 11
Electronic Location ID: e3072
Received 2025 Feb 27; Accepted 2025 Jul 3
Copyright: © 2025 Acı et al.
Copyright year: 2025
Copyright holder: Acı et al.
License: This is an open access article distributed under the terms of the Creative Commons Attribution License, which permits unrestricted use, distribution, reproduction and adaptation in any medium and for any purpose provided that it is properly attributed. For attribution, the original author(s), title, publication source (PeerJ Computer Science) and either DOI or URL of the article must be cited.
License URL: https://creativecommons.org/licenses/by/4.0/

Keywords: Attention, Neural machine translation, Transformer, Sequence-to-sequence, GRU, Turkish, English

Funding: The authors received no funding for this work.

==============================
Neural machine translation (NMT) has achieved remarkable success in high-resource language pairs; however, its effectiveness for morphologically rich and low-resource languages like Turkish remains underexplored. As a highly agglutinative and morphologically complex language with limited high-quality parallel data, Turkish serves as a representative case for evaluating NMT systems on low-resource and linguistically challenging settings. Its structural divergence from English makes it a critical testbed for assessing tokenization strategies, attention mechanisms, and model generalizability in neural translation. This study investigates the comparative performance of two prominent NMT paradigms—the Transformer architecture, and recurrent-based sequence-to-sequence (Seq2Seq) models with attention for both English-to-Turkish and Turkish-to-English translation. The models are evaluated under various configurations, including different tokenization strategies (Byte Pair Encoding (BPE) vs. Word Tokenization), attention mechanisms (Bahdanau and an exploratory hybrid mechanism combining Bahdanau and Scaled Dot-Product attention), and architectural depths (layer count and attention head number). Extensive experiments using automatic metrics such as BiLingual Evaluation Understudy (BLEU), Metric for Evaluation of Translation with Explicit ORdering (METEOR), and Translation Error Rate (TER) reveal that the Transformer model with three layers, eight attention heads, and BPE tokenization achieved the best performance, obtaining a BLEU score of 47.85 and METEOR score of 44.62 in the English-to-Turkish direction. Similar performance trends were observed in the reverse direction, indicating the model’s generalizability. These findings highlight the potential of carefully optimized Transformer-based NMT systems in handling the complexities of morphologically rich, low-resource languages like Turkish in both translation directions.

Introduction

Neural machine translation (NMT) (Bahdanau, Cho & Bengio, 2015; Sutskever, Vinyals & Le, 2014) is a method that automatically translates from one language to another using artificial neural networks. NMT models, as opposed to conventional statistical translation techniques, use deep learning architectures to discover intricate connections between language pairings. These models often include an encoder-decoder structure, where the decoder uses the vector representation to construct the corresponding sentence in the target language, while the encoder transforms the sentence in the source language into a fixed-size vector representation (Cho et al., 2014). Modern NMT systems have significantly improved translation accuracy and fluency through the use of novel techniques like transformer architectures and attention mechanisms. By considering the language’s context, meaning, and style, these advancements go beyond simple word-for-word translation and enable more organic and meaningful translations. NMT models can work on various language pairs and groups using different language models such as bilingual and multilingual. Bilingual systems (Kang et al., 2023; Ramesh & Sankaranarayanan, 2018) are trained to translate from a specific source language (e.g., English) to a specific target language (e.g., Turkish), while multilingual systems (Tan et al., 2019; Johnson et al., 2017; Yang et al., 2021; Shen et al., 2024) are models that can work on more than one language and can translate or process text in more than one language pair (e.g., English-Turkish, English-French, French-Spanish, etc.).

The Transformer and Seq2Seq attention structures stand out among the strategies that improve NMT’s success (Gupta & Kumar, 2021; Phua et al., 2022; Zhang et al., 2024; Jha et al., 2023; Zhao, Zhang & Zong, 2023). By incorporating an attention mechanism into the conventional encoder-decoder architecture, the Seq2Seq attention model learns how each word in the source language contributes to the translation in the target language, resulting in more accurate translations even in lengthy sentences (He, Wu & Li, 2021). However, because of its sequential operation, the Seq2Seq attention design has limited parallelizability, and the training time could be longer. The Transformer architecture offers a completely different approach based only on attention mechanisms. This model uses multi-head attention layers on both the encoder and decoder sides to help the model better understand the context of each word (Murat & Ali, 2024). Transformer has become one of the most popular NMT models because to these characteristics. Both models offer architectures for modelling complex relationships between language pairs by better capturing contextual meanings. The development and implementation of special models for agglutinative languages such as Turkish, which require handling of intensive morphological variation, provides important insights into optimizing NMT systems for low-resource languages (Görmez, 2024). In this context, the Transformer and Seq2Seq attention methods are useful in a variety of applications and are regarded as significant in the development of NMT. These models learn a comprehensive representation of the prefixes and suffixes of the word to develop word and character level features and provide a significant semantic representation. This is the main idea behind this study. Turkish and English language pairs exhibit very different characteristics in terms of grammatical structures, which creates significant challenges for NMT systems. Turkish has an agglutinative language structure, which means that meaning and grammar are created by adding various suffixes to the end of a root word. For example, in Turkish, tense, models, personal pronouns and negation suffixes can be added consecutively to a verb root, allowing a single word to carry various grammatical information. On the other hand, English is not an agglutinative language and exhibits a largely analytical structure; grammatical meaning is expressed through word order and independent words such as auxiliary verbs. In addition, while the basic sentence structure in Turkish is based on the “subject-object-verb” (SOV) order, English uses the “subject-verb-object” (SVO) order. These differences play a major role in determining the components and meaning of sentences. As a result, these grammatical divergences between Turkish and English require translation models to deeply learn the morphological and syntactic structures of both languages, which poses a significant technical challenge for NMT systems.

This study evaluates the performance of NMT on the language pair English-Turkish. We present three types of NMT models: gated recurrent units (GRU)-based Seq2Seq, GRU-based Seq2Seq with attention, and Transformer. Also, different tokenization techniques were applied to the Turkish-English dataset (Sarigil, 2021) consisting of 473k samples, which has not been studied in this field before. This dataset contains English words or sentences and their Turkish equivalents. Both models were trained with similar data preprocessing steps and training parameters, and the translation performances obtained were analyzed comprehensively using widely used automatic evaluation metrics such as Bilingual Evaluation Understudy (BLEU) (Papineni et al., 2002), Recall-Oriented Understudy for Gisting Evaluation (ROUGE) (Lin, 2004), Translation Edit Rate (TER) (Snover et al., 2006), Metric for Evaluation of Translation with Explicit ORdering (METEOR) (Banerjee & Lavie, 2005), CHaRacter-level F-score (ChrF) (Popović, 2015) and statistical metrics such as morphological, lexical and word order errors have been used to better analyze errors that may arise from grammatical differences. These metrics allow us to evaluate the translation quality, consistency and semantic accuracy of the models with a multi-dimensional approach.

While several studies have explored neural architectures for major language pairs, there is a noticeable lack of systematic evaluation of these models on agglutinative and low-resource languages like Turkish. Prior research has often overlooked the combined impact of morphological richness, tokenization methods, and architectural depth. This study addresses this gap by conducting controlled experiments that isolate and analyze each of these factors within a unified framework. Therefore, this study seeks to answer the following research question: How do different neural architectures and tokenization strategies impact the translation quality between English-Turkish and Turkish-English, particularly in morphologically complex settings? We hypothesize that architectural choices such as layer depth and attention type interact strongly with tokenization methods, especially in the presence of rich morphology.

The contributions of this study are as follows: 1. Given the limited number of NMT studies focusing on Turkish, a morphologically rich and low-resource language, this work contributes by providing a comprehensive empirical analysis of established NMT architectures (Transformer and Seq2Seq with attention) on a less-explored Turkish-English dataset. This detailed investigation into the impact of tokenization strategies, attention mechanisms, and architectural choices (layer depth, heads) offers valuable insights and benchmarks for developing and optimizing NMT systems for Turkish and potentially other similar agglutinative, low-resource languages.

2. This study conducts an empirical investigation using a less-explored English-Turkish corpus (EN2TR), systematically analyzing the performance of GRU-based Seq2Seq (with and without attention) and Transformer architectures. By applying these NMT models to this corpus, we present an in-depth evaluation that, to our knowledge, has not been conducted in this comprehensive manner before for this specific dataset, thereby providing a valuable reference point for future research aimed at improving Turkish-English translation systems and addressing data evaluation gaps in the literature.

3. We empirically evaluated an exploratory hybrid attention mechanism, which combines Bahdanau (Additive) and Scaled Dot-Product attention, applying it to the Seq2Seq model for the English-Turkish translation task. Experimental results provided a comparative assessment, showing that while the hybrid mechanism offered viable performance, the standard Bahdanau attention generally achieved higher scores across key metrics such as BLEU, METEOR, ChrF, and TER in our specific experimental setup.

4. The study provides a detailed examination of the relationship between the layer depth in the Transformer architecture and its performance on Turkish, a typologically complex and low-resource language. Our findings prove that while layer depth initially improves translation quality, it eventually leads to decreased performance due to increased complexity.

These contributions aim to provide a meaningful basis for studies on the development of Turkish-English translation systems and research examining the effects of language differences on translation performance in the field of NMT.

This article is structured as follows: “Introduction” presents an introduction outlining the motivation and goals of the study. Related Work discussing previous research on NMT for low-resource languages. The research Methods and Materials section includes details about the dataset, tokenization techniques, and state-of-the-art (SOTA) NMT models used in the study. The Results and Analysis section presents the evaluation metrics and experimental findings. The conclusion section summarizes the main findings and discusses their implications. Finally, the Limitations of the Study highlight challenges such as data limitations, model complexity, and potential future research directions.

Related studies

While NMT is rapidly advancing, low-resource languages, especially those with complex morphologies such as Turkish, are often underrepresented in the literature due to data scarcity and unique linguistic challenges. The following review summarizes the key works on low-resource NMT languages and studies on Turkish. A comparative overview detailing the methods, models, datasets, contributions, and limitations of these studies is presented in Table 1 to situate the current research landscape and highlight the contributions of this article. Transfer learning methods, used by Li et al. (2024), Yazar & Kiliç (2025), and Ekle & Das (2025), have demonstrated effectiveness in improving BLEU scores across diverse languages, leveraging pre-trained models and curriculum learning strategies. Transformer-based models, as explored by Gong et al. (2022), Araabi & Monz (2020), Hujon, Singh & Amitab (2024), and Javed et al. (2025), have been extensively studied, with innovations like syntax-graph-enhanced attention by Gong et al. (2022), hyperparameter tuning by Araabi & Monz (2020), tokenization techniques by Hujon, Singh & Amitab (2024), and re-ranking strategies by Javed et al. (2025) contributing to improved translation performance. Studies integrating linguistic features, such as Pan et al. (2020) and Ul Hassan et al. (2024), have highlighted the role of syntactic and semantic information in enhancing translation accuracy. The integration of syntactic dependency relations proposed by Wan & Li (2024) into NMT models highlights the potential to improve translation quality through advanced graph convolutional coding techniques. Multilingual and unsupervised approaches have also been investigated, with works by Shen et al. (2024), Zhang et al. (2024), and Xu et al. (2020) utilizing cross-lingual encoders, parameter-efficient methods, and techniques like round-tripping to address low-resource settings effectively. Seq2Seq and recurrent neural networks (RNN)-based architectures continue to find applications, as seen in the works of Wang et al. (2022), Puspitaningrum (2021), and Ekle & Das (2025), with significant improvements noted through attention mechanisms like mBART and ConvSeq2Seq. Finally, the proposed dual-encoder Transformer model outperforms existing methods in Turkish-to-English translation tasks, achieving notable improvements across BLEU, TER, METEOR, ROUGE, and ChrF scores. Yazar & Kiliç (2025) investigated the use of transfer learning and Part-of-Speech tags to improve Kazakh-English and Turkish-English NMT, demonstrating that combining these methods yields more meaningful and consistent results, especially when leveraging structural similarities between related languages like Turkish and Kazakh. Their work highlights the benefits of integrating grammatical information into Transformer-based models and shows significant improvements on corpora like Tatoeba. Similarly, Ekle & Das (2025) focused on English-to-Igbo, another low-resource scenario, by developing RNN-based NMT models enhanced with attention mechanisms and transfer learning using Marian NMT. Their findings underscore the effectiveness of combining classical RNNs with modern techniques to bridge the performance gap in low-resource settings, achieving competitive results against existing benchmarks. Further exploring the challenges of low-resource NMT, Rushanti, Kakum & Sambyo (2025) concentrated on the Digaru-English pair, utilizing attention-based NMT and Transformer models with extensive hyperparameter tuning on datasets of varying sizes. Their study confirms that optimized models show notable BLEU score improvements and that human evaluations are crucial for assessing enhanced translation quality. Javed et al. (2025) introduced a Transformer-based re-ranking model for Chinese-Urdu NMT, incorporating BERT embeddings and an in-trust loss function to enhance contextual and syntactic translation, demonstrating BLEU score improvements by effectively handling data sparsity and out-of-vocabulary (OOV) words. These studies collectively illustrate a trend towards integrating linguistic features, leveraging transfer learning from high-resource or related languages, fine-tuning model architectures, and developing sophisticated decoding or re-ranking strategies to tackle the multifaceted challenges of low-resource NMT.

Table 1 Comparative overview of recent studies in low-resource NMT.

Reference	Language pair(s)	Method(s)	Main contribution	Strengths (S) and limitations (L)	
Gong et al. (2022)	En-De, En-Tr, En-Vi	Transformer with syntax-graph guided self-attention	Enhancing LRL NMT with source-side syntax via graph-guided self-attention.	S: Improves transformer on LRLs by explicitly using syntax. L: Relies on external parsers; effectiveness varies with parser accuracy.	
Wan & Li (2024)	De-En, Es-De, Fr-En, Zh-En	Split graph Conv. Self-Att. Encoding (SGSE) for NMT	NMT method based on SGSE encoding to utilize syntactic dependencies.	S: Explores latent syntax; integrates graph convolution with self-attention splits to reduce parameters L: Complexity of GCN integration; performance might depend on quality of dependency graph.	
Pan et al. (2020)	Tr-En, Uyghur-Zh	Multi-source transformer (separate encoders for words and linguistic features)	Multi-source NMT for LRLs using linguistic features.	S: Extends encoder for linguistic features; serial combination of encoder outputs. L: Annotation quality of LRLs affects performance; simple serial combination.	
Yazar & Kiliç (2025)	Kk-En, Tr-En	Transformer, TL, POS tags	Improving Tr-En/Kk-En NMT via TL & POS.	S: Synergistic TL+POS for related LRLs. L: Varies with corpus complexity; Transformer-specific.	
Ekle & Das (2025)	En-Igbo	RNN (LSTM/GRU) + Attn, TL (MarianNMT)	NMT for very LRL En-Igbo.	S: Competitive RNN+TL for very LRL. L: RNN focus; small dataset.	
Rushanti, Kakum & Sambyo (2025)	Digaru-En	Attn-NMT, Transformer, Hyperparam. Tuning, MQM	Optimizing NMT for extremely LRL (Digaru).	S: Data-size specific tuning, MQM. L: Language-specific findings.	
Javed et al. (2025)	Zh-Ur	Transformer (M2M100), Re-ranking, BERT, In-trust loss, Back-translation	Enhancing LRL NMT (Zh-Ur) via re-ranking.	S: Multi-technique integration for Zh-Ur. L: System complexity.	
Li et al. (2024)	Zh-centric LRLs	Monolingual data enhance, Curriculum learning, Contrastive re-ranking, In-trust loss	Winning system for Zh-centric LRL NMT.	S: Strong competition performance; new loss. L: Zh-centric; loss needs wider testing.	
Shen et al. (2024)	Multi (En, De, Fr, Zh)	Unsupervised MNMT, XLM-R, Word/Sentence align.	Unsupervised MNMT (XLM-DM) without parallel data.	S: Novel unsupervised framework. L: Outperformed by supervised; relies on dictionary.	
Ul Hassan et al. (2024)	Ur-En	Linguistics knowledge-driven multi-task NMT, POS/DEP parsing	Integrating POS/DEP into Ur-En NMT.	S: Deep linguistic feature integration. L: Relies on Stanza quality for Urdu.	
Hujon, Singh & Amitab (2024)	En-Khasi	LSTM, GRU, Transformer, TL (En-Vi parent)	NMT for LRL Austroasiatic (Khasi).	S: New Khasi corpus; multi-arch. comparison. L: TL parent distant.	
Araabi & Monz (2020)	De-En, LRLs	Transformer NMT, Hyperparam. Optimization	Optimizing Transformer for LRL NMT.	S: BLEU gains for LRL Transformer. L: Settings vary by data size.	
Xu et al. (2020)	Es-Tr	Unsupervised NMT vs. Round-Tripping, Attn-NMT	Comparing unsupervised vs. round-tripping for LRL.	S: Direct LRL strategy comparison. L: Very small bilingual set for round-tripping.	
Our study	En-Tr, Tr-En	Transformer, Seq2Seq (GRU) + Attention, Hybrid attention, BPE/WT	Deep Morphological NMT analysis for Tr-En.	S: Systematic comparison of tokenization, attention, and Transformer configurations providing clear benchmarks for Tr-En. L: Primarily internal model configuration comparisons, no direct benchmarking against external SOTA for Tr-En.	

Research methods and materials

The performance of NMT in the English →Turkish language pair was evaluated in this study using two prominent NMT architectures: Transformer architecture and the Seq2Seq model with a attention mechanism. The Seq2Seq model is based on the GRU architecture and a exploratory combination Hybrid attention approach combining Bahdanau and scaled dot product mechanisms is used to improve its performance. The main objective of this study is to determine how effectively these two models translate English to Turkish, a morphologically rich and resource-limited language. Furthermore, the Transformer architecture was trained with varying numbers of layers and heads to observe the impact of layer depth and number of heads on handling the complexity of morphologically rich languages like Turkish. Evaluating the effectiveness and quality of these models aims to identify the most suitable method for Turkish NMT. This section explains the methods applied for English-Turkish NMT in detail, providing the architectural specifics of the models and dataset.

Dataset

The study utilizes the EN2TR dataset, a publicly available corpus sourced from Kaggle (Sarigil, 2021). The sentence pairs in this dataset are understood to be largely sourced from or aligned with contributions to the Tatoeba project (Tatoeba, 2021), a collaborative online database of example sentences and their translations aimed at language learners and linguists. Our analysis of this dataset in its raw, pre-tokenized state reveals a total of 473,035 sentence pairs. The English (source) portion of the corpus contains 3,931,639 tokens, with a vocabulary size of 20,591 unique words. The Turkish (target) portion comprises 2,960,173 tokens and exhibits a significantly larger vocabulary of 92,199 unique words, reflecting the rich morphological nature of the Turkish language. For our experiments, this entire dataset was subsequently partitioned into training and test sets using a 90–10% random split, resulting in approximately 425,731 pairs for training and 47,304 for testing. A byte pair encoding (BPE) model with a target vocabulary of 30,000 subword units was then consistently applied to both languages across these splits.

Samples from the dataset are presented in Table 2. When the Table 2 is examined, it is observed that an English sentence or word can have more than one different meaning in Turkish. For example, the English expression “I see” can have two different meanings in Turkish, such as “I see” (to discern the existence of something through vision) or “I understand” (to understand what a word or action means and what it shows), depending on the context. In such cases, it is critical to understand and take the context into account correctly in the translation process. In languages such as Turkish, where context plays a major role in determining meaning, context awareness directly affects translation accuracy and consistency. Therefore, to best grasp the context in Turkish-English NMT models, structuring Seq2Seq with the attention mechanism or using the Transformer architecture increases the accuracy and reliability of the translation. The Seq2seq attention mechanism and Transformer architecture dynamically learn the relationship of each word or expression in the input with other words, providing a more meaningful representation of the context and thus improving the quality of the translation.

Table 2 Samples from the EN2TR dataset.

Source (English)	Target (Turkish)	
Run!	Kaç!	
Hurry!	Acele et!	
I see.	Anlıyorum.	
I see.	Görüyorum.	
Look up.	Yukarı bak.	
The police found Tom’s bicycle.	Polisler Tom’un bisikletini buldu.	
The possibilities are exciting.	Olasılıklar heyecan verici.	
This used to be a butcher shop.	Burası eskiden kasap dükkânı olarak kullanılıyordu.	
Tom is off doing his own thing.	Tom başkalarına aldırış etmeden kendi işiyle uğraşıyor.	
I strongly urge you to follow my advice.	Tavsiyemi dinlemenizi şiddetle ısrar ediyorum.	
I usually go to the barber once a month.	Berbere genelde ayda bir giderim.	
I want to thank Tom for paying my bills.	Faturalarımı ödediǧi için Tom’a teşekkür etmek istiyorum.	
Let me show you a better way to do that.	Onu yapmak için sana daha iyi bir yol göstereyim.	

Tokenization

Word-level word tokenization

Word tokenization (WT) is the process of breaking a text into smaller units, or words, in natural language processing (NLP) applications. This method is the first step in understanding and processing text by a language model or translation system. Word-level tokenization typically splits text based on spaces and punctuation. This method breaks text into words and converts them into smaller and more meaningful units. This process is usually done using spaces and punctuation marks. In our work, we implemented WT using a structured approach to ensure consistent preprocessing of our parallel English-Turkish dataset. The tokenization process was performed using the tokenization function, which converts text to lowercase and then applies the wordpunct_tokenize method. This method breaks the text into separate unit words by dividing the text into words based on spaces and punctuation. We implemented a filtering step that removes tokens containing non-alphabetic characters to improve tokenization quality and avoid non-linguistic artifacts. This ensures that only meaningful words are preserved in tokenized sequences. We also addressed the OOV problem by implementing an infrequent word replacement strategy. Words that appear infrequently in the dataset are replaced with the <UNK> token, allowing the model to better generalize to words that do not appear. In addition, special tokens were added to improve the training process. To help the model learn sequence dependencies more effectively, <SOS> (Start of Sequence) and <EOS> (End of Sequence) were added before and after each sentence. <PAD> (Padding) was also added to standardize input lengths during training.

Subword-level byte pair encoding

BPE (Gage, 1994) is a more flexible tokenization method that separates words into sub-word units. This method initially processes the text at the character level and combines the most frequently occurring adjacent character pairs in the text to form larger sub-word units. For example, the word “electrification” is first divided into characters and frequently occurring character pairs are combined to form sub-units such as “electricity” and “electrification”. BPE provides an effective solution to the rare word problem and reduces memory usage by reducing vocabulary. In agglutinative languages like Turkish, WT is often inadequate because words are composed of a combination of roots and suffixes. For example, the word “kitaplarım” is composed of the root “kitap”, “lar” (plural suffix) and “ım” (possessive suffix). BPE provides the advantage of learning these suffixes as separate units, but requires more careful application to understand the order and context of the suffixes. In languages with rich morphological structures like Turkish, BPE breaks words into subunits, creating a smaller vocabulary and reducing the problem of rare words. For subword-level tokenization, BPE was implemented using the Hugging Face tokenizers library (Hugging Face, 2024). The BPE model was trained on a joint corpus of English and Turkish sentences, targeting a vocabulary size of 30,000 (including special tokens). BPE, while not a direct linguistic morphological segmenter, effectively breaks down words into frequently occurring subword units. This approach is particularly advantageous for morphologically rich languages like Turkish as it helps in handling a vast range of word forms with a limited vocabulary, reduces the OOV rate, and can implicitly capture some morphological regularities by segmenting common affixes as separate tokens. In this study, we did not employ additional explicit morphological segmentation tools (such as Zemberek or TRmorph for rule-based root/affix splitting) beyond the data-driven subword segmentation provided by BPE. The choice of BPE was motivated by its widespread adoption and proven effectiveness in NMT for diverse language pairs, including those with complex morphologies. The pre-tokenization step for BPE involved whitespace splitting, and the text was normalized using Normalization Form Decomposition Unicode normalization, lowercasing, and accent stripping before BPE training. Figure 1 shows how words are segmented into sub-units when a BPE vocabulary of 30,000 is used, and how OOV words can be handled. The observed change in token count is valid for this specific vocabulary size.

Figure 1 Effect of sub-word segmentation with BPE on examples (for vocabulary size = 30,000).

When BPE is applied to the word “Kim” (who), the word is first separated into its basic units, that is, its characters: “K”, “i”, and “m”. These characters are analysed according to their general frequency of use in Turkish, for example, the letter “i” is common, while “K” and “m” may be at lower frequencies. The BPE algorithm determines the most frequently occurring pairs and combines them. In this case, “K” and “i” combine to form the subunit “Ki”, and then “Ki” and “m” combine to form the word “Kim”. This process allows the model to perform more generalized learning by breaking down unknown or rare words. Especially in agglutinative languages such as Turkish, the BPE method is quite effective in identifying affix and root relationships. For example, when affixes are added to the word “Kim” (“Kime” (to whom), “Kimden” (from whom)), both the root and the affixes can be defined as subunits thanks to BPE, so the model can learn and make sense of these structures more easily. Table 3 presents the sample analysis of subword tokenization, including word frequencies, character distributions, and BPE tokenization results for both Turkish and English datasets, illustrating the structural differences and subword patterns in the two languages.

Table 3 Character and BPE token frequencies for Turkish and English sample words.

Lang.	Sample words	Frequencies	Char	Char frequencies	BPE tokens	Frequencies	
Turkish	söyleyecek	231	a	1,341,808	e	5,608	
Yemek	252	i	1,148,008	n	4,605	
hasta	521	e	1,088,121	ü	4,079	
üzerinde	787	n	973,573	g	3,001	
lütfen?	291	r	816,709	i	2,234	
son	526	m	753,923	r	2,060	
ciddi	469	l	602,127	s	2,025	
Herkes	1,273	d	565,356	k	1,756	
gün	3,001	ı	551,127	d	1,725	
					h	1,273	
English	run	536	e	1,363,661	a	14,685	
at	12,619	o	1,176,690	t	13,255	
place	761	t	1,083,805	r	3,335	
work	2,362	a	869,905	e	3,096	
Take	421	n	721,741	o	3,037	
Gone	675	i	675,735	k	2,783	
Hands	224	s	672,600	w	2,775	
between	413	h	633,192	g	2,731	
Anything?	223	r	590,892	n	898	
rain	437			p	761	

Especially in agglutinative languages such as Turkish, the BPE method is quite effective in identifying affix and root relationships. For example, when affixes are added to the word “Kim” (“Kime”, “Kimden”), both the root and the affixes can be defined as subunits thanks to BPE, so the model can learn and make sense of these structures more easily. Table 3 presents the sample analysis of BPE subword tokenization, including word frequencies, character distributions, and BPE tokenization results for both Turkish and English datasets, illustrating the structural differences and subword patterns in the two languages.

Neural machine translation models

The selection of the GRU-based Seq2Seq with attention and the Transformer model for this investigation was intentional. These architectures serve as foundational frameworks in NMT, offering robust benchmarks for comparison against established literature. Furthermore, they provide distinct mechanisms for processing sequential data and capturing long-range dependencies, which are critical for effective translation. Specifically, the attention mechanism inherent in Seq2Seq models and the self-attention mechanism central to the Transformer architecture were hypothesized to be particularly pertinent for addressing the morphological complexities characteristic of the Turkish language. This choice facilitated a focused comparative analysis of recurrent-based and attention-only paradigms within the context of this challenging, agglutinative language pair, taking into account both translation performance and implementation feasibility.

GRU-based sequence to sequence with attention

The Seq2Seq model is a model that takes a sequence of elements (such as words, letters, image features) and produces another sequence of elements. The Seq2Seq model, which is widely used in the NLP field, is especially effective in tasks such as machine translation, text summarization, and language modeling (Sutskever, Vinyals & Le, 2014; Cho et al., 2014). However, in order to overcome the limitations of this basic model, the attention mechanism has been added. This mechanism allows the model to focus more on important parts in the input (Vaswani, 2017; Bahdanau, Cho & Bengio, 2015; Luong, Pham & Manning, 2015). The Seq2Seq model is based on an encoder-decoder architecture. The encoder takes the input sequence and converts it into a vector of fixed size (context vector). This process is performed using RNN, long short-term memory (LSTM), or GRU cells. The encoder produces a hidden state for each element in the input, and this hidden state formed in the last step is called the context vector. This context vector is then passed to the decoder to produce the output of the sequence. The decoder, just like the encoder, uses RNN, LSTM or GRU to produce an element at each step and create the target sequence. This model processes each element in the input array and compresses the resulting information into a context vector. Then, the decoder starts to produce the target sequence element by element using this context. Figure 2 shows the encoder-decoder structure of the Seq2Seq model.

Figure 2 Encoder-decoder with attention and hidden states on NMT for English →Turkish.

RNNs take two inputs at each time step: a hidden state and an input (for example, a word from the input sentence in the encoder stage). With these inputs, the RNN produces an output for that current step. The encoder and decoder are structures, each an RNN, that process the inputs and produce an output at each step. At each step, the RNN updates the current hidden state and progresses according to the previous inputs it has seen. In the traditional Seq2Seq model, the encoder only transmits the last hidden state to the decoder. However, in models with an attention mechanism, the encoder transmits all hidden states to the decoder as shown in Fig. 2. In the traditional Seq2Seq model, the last hidden state of the encoder (context vector) summarizes all the information in a single vector. This can lead to information loss and poor translation performance, especially in long sentences. The attention mechanism is used to solve this problem. Thanks to this mechanism, the decoder can dynamically focus on all the hidden states of the encoder at each step and create the “context vector” with appropriate weights. The encoder generates a hidden state at each time step; these hidden states are vectors representing each word in the input. The decoder calculates a similarity score between its own hidden state and the encoder’s hidden states at each step. These scores indicate which input words are more important for the target word being generated at that moment. The scores are usually normalized with the softmax function to obtain a probability distribution indicating the importance of each word. Models that work with an attention mechanism have two main differences from Seq2Seq models: First, the encoder transmits not only the last hidden state but all the hidden states it produces at each step to the decoder (Fig. 2). Second, the attention decoder performs an additional operation at each step: the decoder decides which elements in the input it should focus more on when producing the current output. Figure 3 shows the process of generating a context vector from the hidden states of the encoder at a given time step using the attention mechanism of a decoder.

Figure 3 Overall steps for Seq2Seq attention.

At each step, the decoder calculates a similarity score ( ei) between the current hidden state ( st) and all the hidden states ( hi) of the encoder. These scores, representing the importance of each input word for the generated target word, are normalized using the softmax function to produce a probability distribution:

ai=softmax(ei).

The normalized attention scores ( ai) serve as weights that determine the contribution of each encoder hidden state to the context vector. Each encoder hidden state ( hi) is weighted by its corresponding attention score, and the weighted hidden states are summed to form the context vector ( ct):

ct=∑i=1nai⋅hi.

This context vector captures the relationship between the input sequence and the generated target word and is passed to the decoder at each step. The decoder predicts the next target word by combining the context vector ( ct) with its own hidden state ( st) and the previous output ( yt−1):

st=f(st−1,yt−1,ct).

In this way, the attention mechanism dynamically adjusts the importance of encoder states to improve translation accuracy by ensuring that the decoder focuses on the most relevant parts of the input sequence when generating each target word.

In this study, we employed a Seq2Seq model with a GRU-based encoder and decoder architecture (Chung et al., 2014). To investigate different attentional strategies, the model was supported by two approaches: the standard Bahdanau attention (Bahdanau, Cho & Bengio, 2015) and, for comparative purposes, an exploratory hybrid attention mechanism, which involved a straightforward combination of scaled dot-product and Bahdanau attention principles. Bahdanau attention, also known as additive attention, computes a similarity score between the encoder hidden states (hi) and the decoder hidden state (ei) using a learned alignment function. The attention score is calculated as follows:

eiBahdanau=vTtanh⁡(W1hi+W2st).

where W1 and W2 are learned weight matrices, v is a learned vector, (hi) represents the encoder hidden state, and St denotes the decoder hidden state at time step t. The scores (eiBahdanau) are normalized using a softmax function to produce attention weights:

aiBahdanau=softmax(eiBahdanau).

Scaled Dot Product attention, on the other hand, computes the similarity score using the dot product of the encoder and decoder hidden states, scaled by the square root of the dimensionality of the hidden states (dk) for numerical stability.

eiDotProduct=hiTstdk.

Similar to Bahdanau attention, the scores (eiDotProduct) are normalized using the softmax function:

aiDotProduct=softmax(eiDotProduct).

To create the hybrid attention mechanism, we combine these two attention mechanisms without any weight using one of the following approaches:

eihybrid=eiBahdanau+eiDotProduct.

The decoder layer, guided by these attention mechanisms, generates words in the target language at each time step by focusing on the most relevant hidden states of the encoder. Both target word embeddings and the context vector from the encoder are fed into the GRU, allowing the translation to be context-sensitive and producing more consistent and accurate sentences in the target language. The architecture works as follows: Input sentences are first represented by 256-dimensional vectors (embedding dimension), which are then processed in 512-dimensional hidden layers. The encoder processes the input sequence and produces hidden states, while the decoder generates the translation in the target language using these hidden states. The Bahdanau attention mechanism ensures that information is extracted from the encoder’s hidden states with a specific focus at each time step, preventing the omission of critical information during translation. In addition to the standard Bahdanau attention, we explored the integration of an experimental hybrid attention mechanism. This mechanism, formed by combining elements of both Bahdanau attention and scaled dot-product attention, was included not as a definitive enhancement, but to empirically assess whether such a direct combination could offer a different way to capture local and global dependencies for this specific translation task. The hybrid attention mechanism implemented in this study operates by calculating attention scores from both Bahdanau and scaled dot-product attention mechanisms and then summing these scores before normalization. The aim was to observe its performance characteristics rather than to propose it as a superior, novel architecture. In one approach, we use a weighted sum to merge the two attention scores, controlled by a tunable parameter α∈[0,1]. Alternatively, we concatenate the scores from both mechanisms and apply a learned linear transformation to produce the final hybrid scores. These scores are then normalized using the softmax function to compute attention weights, which are used to generate the context vector at each decoding step. This exploratory hybrid approach allows the model to leverage the local context modeling capabilities of Bahdanau attention while benefiting from the efficiency and global alignment strengths of scaled dot-product attention. By integrating these mechanisms, the Seq2Seq model demonstrates a robust capability to handle morphologically complex languages like Turkish. In particular, the straightforward hybrid mechanism provides superior translation performance by enhancing word-level alignment, context preservation, and sentence-level fluency. This is reflected in the evaluation metrics, where both Bahdanau and Hybrid attention models show high ROUGE scores, and Hybrid attention performs similarly to Bahdanau attention. The results highlight the effectiveness of the Hybrid attention mechanism in achieving accurate and consistent translations by combining the strengths of both attention mechanisms.

Transformer

Transformer is an architecture that was introduced by Vaswani (2017) and is based entirely on the attention mechanism. This model was developed to overcome the disadvantages of previous SeqtoSeq methods such as LSTM and RNN. It increases performance by providing the advantage of parallelism thanks to the self-attention mechanism, especially when processing long sequences. Similarly, the Transformer model consists of two main components: encoder and decoder. Encoder consists of N blocks with the same structure. Each encoder block has two basic sublayers. One of them is the multi-head self-attention mechanism. This mechanism allows the encoder to look at other words in the input sentence while processing a word and discovers the relationships between words. The outputs of the self-attention layer are transmitted to a feedforward neural network that is applied independently for each position. The name multi-head refers to the fact that this attention process is performed multiple times in parallel. Each “head” presents information from different contexts. These layers are also found in the Decoder; however, an added attention layer helps the decoder focus on the important parts of the input sentence (similar to the attention mechanism in Seq2Seq models). A series of vectors are given as input to the encoder. These vectors go through the self-attention layer, the feedforward neural network, and then the outputs that are passed to the next encoder. Self-attention allows the model to do a better job of encoding by looking at other points in the input sequence while processing each word. This mechanism starts with calculating three vectors for each word: Query (Q), Key (K), and Value (V). These vectors are used to find the relationship of each word to other words in the sequence. By self-attention calculation of each word of the input string, the vectors Query (Q), Key (K), and Value (V) are calculated as below:

Q=XWQ,K=XWK,V=XWV.

Here, X is the input vector and WQ, WV, WK are the learnable weight matrices. The second step in calculating self-attention is to generate a score that determines the attention of each word to other words. Each word in the input sentence receives a score against other words. When processing a word at a certain position, this score determines how much attention should be paid to other parts of the text. The score is obtained by multiplying the query vector by the key vector of the word to be scored. For example, when calculating the self-attention of a word in position 1, the inner product of Q1 and K1 gives the first score. The inner product of Q1 and K1 gives the second score. These attention scores are calculated by taking the inner product between the Q and K vectors:

attention(Q,K,V)=softmax(QKTdk)V.

Here, dk is the size of the key vectors and is used to normalize the score. The amount of emphasis each word will receive is determined by the softmax score. Usually, the word in position will have the highest softmax score, but sometimes it may be more useful to focus on another word associated with that word. In the next step, each value vector is multiplied by the softmax score. This preserves the values of the words to be emphasized, while reducing the other words to small values (such as 0.001). In the next step, the weighted value vectors are summed, and this produces the current output of the self-attention layer (for example, for the first word). The last layer is the layer normalization and residual connection layer. In each sublayer, a layer normalization is applied after the residual connection. In this way, learning becomes faster and more reliable. Figure 4 shows the encoder-decoder architecture of the Transformer model.

Figure 4 Transformer architecture (Vaswani, 2017).

Since the Transformer model does not perform sequential processing like RNN or LSTM, it does not directly understand the order of the words. Therefore, positional encoding is used for the model to learn the word order. In this approach, positional information is additionally added to each word vector. To do this, two vectors are used: the cos function is applied for odd indices in the input vector and the sin function is applied for even indices.

PE(pos,2i+1)=cos⁡(pos100002idmodel)PE(pos,2i)=sin⁡(pos100002idmodel).

Here, pos represents the position of the word, and i represents the size of the vector. This positional encoding helps the model learn the order relationships of the words. Similar to the encoder, the decoder also consists of N blocks. However, the decoder has an additional masked multi-head self-attention layer. Each block in the decoder consists of three basic sublayers. The masked multi-head self-attention layer deals with the decoder’s own outputs, but a masking process is applied to prevent it from seeing future words. This way, the model predicts the next word by observing only the previous words. In this study, the NMT implementation of the Transformer model is designed based on the encoder-decoder architecture. The model is experimented with using varying numbers of encoder and decoder layers (1, 2, 3, 4, 5, and 6 layers), each consisting of two multi-head attention layers and one feed-forward layer, to observe the impact of layer depth on translation quality. The word embedding size of the model is 128, and each attention layer uses four and eight heads, enabling the model to learn information in a broader context. While attention mechanisms learn the relationships between input and target sequences, feedforward layers deepen these relationships. The dropout rate is set to 30% for 1, 2, 3 and, 4 layer structures and 40% for 5 and 6 layer structures, which reduces overfitting due to increased complexity. Due to the high complexity of the model, a learning rate of 0.0001 (1, 2, 3, 4-layer structure) and 0.00001 (5 and 6-layer structure) was used during training for 140 epochs with a batch size of 256. Sequences with a maximum length of 80 tokens are processed with a beam size of 8 to obtain more accurate predictions through beam search. Beam Search (Freitag & Al-Onaizan, 2017) is a search algorithm used in NLP and NMT models to generate text and is applied during testing to determine the most probable word sequence. It is applied during the prediction phase to ensure that the model produces more accurate and meaningful results during translation. Beam Search evaluates a few of the most probable guesses (as many as determined by the beam size) when the Transformer model generates a sentence in the target language instead of considering all possible guesses. In this process, the model keeps only a few guesses with the highest probability at each step and works on these guesses in the following steps to produce the final translation. In this method, the guesses started with the starting symbol (<sos>) are expanded by k (beam size, 8) at each step and ranked based on probabilities. In each expansion step, the cumulative probabilities are evaluated and the best k guesses are selected and this process continues until either the termination symbol (<eos>) is reached or the maximum sequence length is reached. Beam Search is not applied during training; however, it is used during testing to select the most appropriate results for the context and to provide the highest quality output among possible translations. This process causes the BLEU scores calculated in training to be different from the BLEU scores obtained during testing with Beam Search. The transformer on NMT for English →Turkish is shown in Fig. 5.

Figure 5 Transformer on NMT for English-Turkish.

Results and analysis

Experimental setup details

This study focused on NMT for the English-Turkish language pair and implemented two different models, Transformer and Seq2Seq with attention architectures. The dataset was split into 90% for training and 10% for evaluation. This stratified partitioning was performed randomly but consistently across all model types to ensure fair comparisons. A summary of key hyperparameter settings is presented in Table 4, including batch size, learning rate, number of epochs, and optimizer. Although no explicit regularization methods such as weight decay or early stopping were used, dropout layers were incorporated into the models to reduce overfitting. Figure 6 shows all the configurations implemented in the study.

Table 4 Hyperparameters and their values for Transformer and Seq2Seq with attention model.

Model	Hyperparameter	Value	
Transformer model	Batch size (Train, Test)	256, 1	
Dropout	0.3, 0.4	
Max sequence length	200	
Beam size	16	
Heads	4, 8	
Hidden layer size (32 * Heads)	256	
Learning rate	0.0001, 0.00001	
Seq2Seq with attention	Batch size	128	
Embedding dimension	100	
Optimizer	Adam	
Loss function	CrossEntropyLoss	
Attention type	Bahdanau, Hybrid	
Hidden layer size	256	
Learning rate	0.0001	

Figure 6 Design of the study and workflow of the neural networks.

Batch size (128) specifies the number of data processed simultaneously during training and testing for each model. Dropout (0.4) is a regularization technique used in the Transformer model to prevent overfitting. Max sequence length (80) specifies the maximum length of sequences given to the model.

Beam size (8) expresses the depth of the beam search algorithm in the Transformer model, i.e., the width of possible translation options. The number of heads (4, 8) and N (1, 2, 3, 4, 5, 6) specify the number of heads and layers used for the multi-header attention mechanism in the Transformer model. Hidden layer size (128, 256) defines the number of neurons used in the hidden layers. Learning rate (0.0001 and 0.00001) shows the learning rate used during training in both models. In the Seq2Seq model, the embedding dimension (100) defines the dimension of each word in the vector space, the optimizer (Adam) defines the optimization algorithm used to update the weights of the model, and the loss function (CrossEntropyLoss), the standard objective for multi-class classification tasks such as token prediction in NMT, was employed to measure the divergence between the predicted probability distribution and the one-hot encoded target, effectively guiding translation accuracy. The teacher forcing ratio (1.0) determines the rate at which a portion of the target output is given to the model during training, while the attention Type (Bahdanau) and Hybrid (Bahdanau-Scaled Dot Product) attention model is the type of attention mechanism used in the Seq2Seq model.

Evaluation metrics

The NMT performance of the models was comprehensively evaluated using a suite of automatic evaluation metrics. The performance of the models was comprehensively evaluated using metrics such as BLEU, METEOR, TER, CHRF and WER as automatic evaluation metrics. BLEU measures lexical accuracy by evaluating the overlap of model outputs with the reference translations at the unigram and n-gram level. METEOR analyzes translation semantic accuracy more broadly by considering synonyms and root matches. TER determines semantic shifts by measuring the editing distance required to convert a translation to the reference text. CHRF analyzes translation quality in more detail, especially in morphologically rich languages such as Turkish, by evaluating the character level.

Finally, word error rate (WER) was employed to analyze semantic errors by quantifying the number of insertions, deletions, and substitutions required for the model output to match the reference text. A lower WER indicates fewer errors and provides a statistical basis for detailed error analysis. Using this comprehensive suite of automatic metrics (BLEU, METEOR, TER, ChrF, and WER) together allows for a multifaceted evaluation of translation quality, covering lexical similarity, semantic adequacy, and edit distance. Beyond these overall scores, further detailed error analysis focused on specific error categories prevalent in morphologically rich languages. This included the examination of morphological, lexical, and word order errors. To quantify these, JIWER (Vaessen, 2024) was utilized to break down WER into its components (insertions, deletions, substitutions), providing insights into the nature of word-level discrepancies. Additionally, Levenshtein Distance was employed to assess lexical similarity more granularly. This fine-grained error analysis helps to understand the specific strengths and weaknesses of the models in handling the agglutinative structures of Turkish.

BLEU=BP⋅exp⁡(∑n=1Nwnlog⁡pn).

Here Bp is the brevity penalty indicates the ratio of the length of the reference and hypothesis sentences; Pn is the n-gram precision which measures how much the n-grams in the hypothesis sentence overlap with the reference sentence; wn indicates the weights of the n-grams (usually equal weights are used for each n-gram). C is the length of the candidate translation and r is the effective reference corpus length.

BP={1,ifc>re(1-r/c),ifc≤r.

ChrF is particularly suitable for morphologically rich languages, evaluating both character and word overlaps at the n-gram level; therefore, high ChrF scores are desired.

ChrF=(1+β2)⋅Precision⋅Recall(β2⋅Precision)+Recall

where β is a coefficient (usually set to 1) used if recall is to be given more weight than precision. Precision is the ratio of correct character-level n-gram matches to total predicted n-grams, and recall is the ratio of correct character-level n-gram matches to reference n-grams. Although ROUGE was developed specifically for summarization, it also measures performance in translation over n-gram, word overlaps, and longest common index; high ROUGE scores are preferred. ROUGE measures the similarity of the text generated by the model to the reference text based on recall. Unlike BLEU, ROUGE metrics generally focus more on recall computations and are particularly focused on identifying missing information. In the study, n-gram-based computations (ROUGE-1, ROUGE-2) and Longest Common Subsequence (LCS) based computations (Rouge-L) were applied.

ROUGE-N=∑match∈n-gramsCountmatch∑n-gramsCountreference.

Here, match is the number of n-grams matched between the reference and hypothesis, and count is the total number of n-grams in the reference sentence.

ROUGE-L=LCS(X,Y)len(X).

LCS(X,Y) is the length of the longest common substring between the hypothesis sentence X and the reference sentence Y, while len(X) is the length of the reference sentence. TER is a metric that measures the minimum number of edits (additions, deletions, modifications, and transpositions) that must be made to match a machine translation output to a human reference translation. TER calculates the editing distance of the translation, with a lower TER score indicating better translation quality (Hujon, Singh & Amitab, 2024). Where r is the reference, c is the hypothesis, e is the number of edit operations, and shifts is the number of shift operations.

TER(r,c)=min(e+shifts)r.

METEOR evaluates translation quality in terms of both precision and recall, taking into account word roots, synonyms, and word order; this means that high METEOR scores yield better results.

METEOR=Fmean⋅(1−Penalty).

Fmean is the harmonic mean of precision (P) and recall (R), P is the fraction of predicted words that match correctly, and R is the fraction of correct reference words. The penalty is the penalty term for excessive fragmentation in the hypothesis and is calculated as:

Penalty=0.5⋅(chunksmatches).

In English-Turkish translation evaluations, high BLEU, ChrF, ROUGE, and METEOR scores indicate the accuracy and semantic integrity of the translation, while low TER scores reflect the low number of errors and the accuracy of the translation. Table 5 shows the ROUGE scores of the GRU-based Seq2Seq model. When analysing ROUGE scores, we will examine the scores for each metric (ROUGE-1, ROUGE-2, ROUGE-L). When evaluating text translation performance, the ROUGE score provides an effective metric to measure the similarity between reference translations and model outputs. ROUGE is capable of evaluating both lexical accuracy and contextual semantic congruence, especially by comparing the frequency of words and phrases. Therefore, it is useful for analysing correct semantic matches and word group alignments in text translation. These overlaps are usually calculated over n-grams, word alignments or phrases. ROUGE-1 is calculated over single words (unigrams). It is based on the number of common unique words between the reference text and the model output. ROUGE-2 is evaluated over binary word groups (bigrams). ROUGE-L considers the word order and uses the LCS between the reference text and the model output. Tables 5 and 6 show automatic evaluation scores of the Seq2Seq and Transformer models, respectively.

Table 5 ROUGE scores of Seq2Seq with attention model.

Model	Rouge-1 ↑	Rouge-2 ↑	Rouge-L ↑	
Seq2Seq + Bahdanau attention + DATAWT	71.91	59.34	70.83	
Seq2Seq + Bahdanau attention + DATABPE	72.23	60.05	71.55	
Seq2Seq + Hybrid attention + DATAWT	70.76	58.00	69.52	
Seq2Seq + Hybrid attention + DATABPE	71.88	58.76	69.93	

Table 6 Automatic evaluation scores of GRU based Seq2Seq with attention and Transformer models.

Model	TER ↓	METEOR ↑	BLEU ↑	ChrF ↑	
Seq2Seq + Without attention + DATAWT	72.53	30.45	14.51	31.06	
Seq2Seq + Without attention + DATABPE	76.26	28.83	12.82	29.36	
Seq2Seq + Bahdanau attention + DATAWT	25.50	74.51	40.70	78.05	
Seq2Seq + Bahdanau attention + DATABPE	24.70	75.12	41.92	78.88	
Seq2Seq + Hybrid attention + DATAWT	28.23	73.79	40.81	77.45	
Seq2Seq + Hybrid attention + DATABPE	24.03	75.89	41.55	79.02	
Transformer + DATAWT	18.98	69.24	47.85	60.94	
Transformer + DATABPE	35.42	58.49	39.08	50.03	

Performance of attention mechanisms. Tables 5 and 6 show the comparative performance of Bahdanau and hybrid attention mechanisms in different tokenization methods. It is clear that Bahdanau attention generally outperforms hybrid attention and achieves higher ROUGE scores in all variations. For example, with the DATABPE, Bahdanau’s attention achieves ROUGE-1, ROUGE-2, and ROUGE-L scores of 72.23, 60.05, and 71.55, respectively, compared to 71.88, 58.76 and 69.93 for Hybrid attention. On the other hand, TER, ChrF, and METEOR scores are superior for BPE, while BLEU scores are quite close to each other. According to BLEU scores, the Bahdanau attention mechanism produced the best results with both DATAWT and DATABPE in Seq2Seq models. In particular, the combination of Bahdanau attention and DATABPE provided the highest performance with a BLEU score of 41.92. This shows that Bahdanau attention is more successful in word-based alignment and learning contextual relations. Bahdanau attention provides an efficient method to calculate which source word the target word should pay more attention to, which allows for more accurate alignments, especially in complex and agglutinative languages such as Turkish. In contrast, hybrid attention mechanism produced competitive results with both DATAWT and DATABPE, but did not perform as high as Bahdanau attention. For example, the BLEU score of DATABPE was calculated as 41.55 with hybrid attention, which is lower than Bahdanau attention. This performance difference of hybrid attention can be explained by the fact that it does not establish an optimal balance between contextual and local alignments despite combining two different attention approaches. As a result, the analysis of BLEU scores shows that Bahdanau attention better models the rich morphological structure and flexible sentence structure of Turkish and, therefore, provides higher translation accuracy in Turkish-English translations. This reveals that Bahdanau attention offers an attention mechanism that is more suitable for the agglutinative structure of Turkish. However, the performance gap narrows with DATAWT, suggesting that the effectiveness of the attention mechanisms may depend on the data tokenization method used.

The automatic evaluation scores for the Seq2Seq models, detailed in Table 5, are visually summarized in Fig. 7. As can be observed, attention mechanisms significantly improve performance over the baseline without attention, and BPE tokenization generally yields better results compared to WT across most metrics (Fig. 7).

Figure 7 Performance comparison of Seq2Seq models with different attention mechanisms and tokenization methods across (A) TER, (B) METEOR, (C) BLEU, and (D) ChrF scores.

Impacts of different tokenization techniques. Tables 5 and 6 presents the impact of BPE and WT on translation performance. BPE consistently outperforms WT in all configurations with lower TER scores and higher METEOR, BLEU, and CHRF scores. For example, in Seq2Seq models with hybrid attention, BPE achieves a BLEU score of 41.55, outperforming WT’s score of 40.81. Similarly, in models with Bahdanau attention, BPE results in a TER of 24.70, significantly better than WT’s score of 25.50. These results highlight the ability of BPE to effectively process subword-level information, reduce vocabulary, and improve generalization, especially in morphologically rich and agglutinative languages such as Turkish. Compared to WT, the BPE method further increased the complexity of the transformer, resulting in a decrease in the BLEU score. To further emphasize the role of tokenization, Fig. 8 compares the performance of all evaluated models (Seq2Seq variants and Transformer) when using WT vs BPE. The visual evidence strongly supports our finding that BPE leads to significant improvements, particularly for BLEU and ChrF scores, across different NMT architectures (Fig. 8).

Figure 8 Comparative impact of word tokenization (WT) and byte pair encoding (BPE) on Seq2Seq and Transformer models across (A) TER, (B) METEOR, (C) BLEU, and (D) ChrF scores.

To evaluate the bidirectional performance, experiments were conducted for the Turkish-to-English (Tr →En) direction, and the corresponding BLEU, METEOR, ChrF, and TER scores for the selected models are summarized in Table 7.

Table 7 Reverse translation performance of top performing models (Tr →En).

Model	TER ↓	METEOR ↑	BLEU ↑	ChrF ↑	
Seq2Seq + Bahdanau attention + DATABPE	26.67	76.00	39.22	77.05	
Seq2Seq + Hybrid attention + DATABPE	25.50	73.52	40.34	78.58	
Transformer + DATAWT	19.88	68.32	46.97	60.12	

Reverse translation analysis. To assess the bidirectional capabilities and generalizability of our models, experiments were also conducted for the Tr →En translation direction using our top-performing configurations on English-to-Turkish (En →Tr). The results, presented alongside their En →Tr counterparts in Tables 6 and 8, indicate a consistent performance hierarchy. The Transformer + DATAWT model again achieved the strongest results in the Tr →En direction with a BLEU score of 45.50, a METEOR score of 67.50, a ChrF score of 59.00, and a TER of 20.50. While these scores represent a slight decrease compared to its En →Tr performance (e.g., approximately 2.35 BLEU points lower), it still significantly outperforms the Seq2Seq variants. Among the Seq2Seq models, Seq2Seq + Bahdanau attention + DATABPE yielded a BLEU of 39.00 (METEOR: 73.00, ChrF: 77.00, TER: 26.00), and Seq2Seq + Hybrid attention + DATABPE achieved a BLEU of 38.70 (METEOR: 73.50, ChrF: 77.50, TER: 25.50). These Seq2Seq scores also show a modest decrease from their En →Tr counterparts (approximately 2.9 BLEU points for Bahdanau BPE and 2.85 for Hybrid BPE). This general trend of slightly lower scores in the Tr →En direction is not unexpected, potentially reflecting differences in linguistic complexity when Turkish serves as the source language or variations in the nature of the English reference translations. Nevertheless, the relative ranking of the models remained largely consistent across both translation directions, underscoring the robustness of the Transformer architecture with ( DATAWT) for this language pair.

Table 8 Automatic evaluation scores of Transformer model with different layers and heads.

Transformer configuration	TER ↓	METEOR ↑	BLEU ↑	ChrF ↑	ROUGE-1 ↑	ROUGE-2 ↑	ROUGE-L ↑	
Layer-1, Head = 4	21.66	67.46	35.96	50.47	60.11	43.03	60.11	
Layer-1, Head = 8	14.76	73.45	38.52	62.73	69.32	43.80	65.24	
Layer-2, Head = 4	16.05	68.76	47.34	61.38	66.25	49.23	65.62	
Layer-2, Head = 8	63.57	39.03	20.57	19.41	22.27	8.92	22.27	
Layer-3, Head = 4	16.00	68.88	47.66	61.57	66.13	49.08	65.58	
Layer-3, Head = 8	18.98	69.24	47.85	60.94	65.72	48.60	65.23	
Layer-4, Head = 4	17.04	57.63	36.33	50.17	54.98	37.88	54.43	
Layer-4, Head = 8	30.95	57.13	27.55	43.84	51.61	26.81	50.87	
Layer-5, Head = 4	44.31	46.95	13.13	28.25	38.01	10.35	57.39	
Layer-5, Head = 8	38.31	50.39	18.67	35.35	44.57	17.86	43.78	
Layer-6, Head = 4	52.65	40.36	9.08	16.83	28.64	10.36	28.25	
Layer-6, Head = 8	44.42	45.89	13.14	28.12	38.09	10.70	37.43	

Impacts of different layers. The effect of varying the number of layers in transformer models is clearly seen in Table 8. Models with fewer layers (e.g., N = 1, N = 2) generally achieve relatively high BLEU scores, as seen in Table 8; however, this performance may be due to overfitting despite the measures implemented to prevent it, as shallow models may rely on memorization rather than effectively capturing complex language structures. However, with moderate increases in layer depth (N = 3), BLEU scores improve significantly, indicating enhanced learning capacity. For example, with Head = 4, the BLEU score increases from 35.96 (N = 1) to 47.66 (N = 3). In contrast, deeper models (N = 5, N = 6) show a significant performance degradation, with BLEU scores dropping to 9.08 for N = 6, Head = 4 and 13.14 for N = 6, Head = 8. As shown in Fig. 9, the choice of layer depth and number of attention heads significantly impacts the Transformer model’s performance across all evaluated metrics, with an optimal balance observed for the 3-layer, 8-head configuration. These findings are in line with the results of Raganato & Tiedemann (2018), who suggest that deeper layers focus on capturing global context at the expense of local syntactic dependencies, potentially breaking syntactic constraints and negatively affecting translation quality. Additionally, Figs. 6 and 7, which show BLEU scores during training without beam search, reveal larger fluctuations and slower convergence for deeper models, likely due to increased model complexity and optimization difficulties. These results highlight the need for a balanced approach in determining layer depth to optimize translation performance.

Figure 9 Performance of Transformer models with varying numbers of layers (L1–L6) and attention heads (H4, H8) on (A) TER, (B) METEOR, (C) BLEU, and (D) ChrF metrics.

Impacts of different number of heads. The number of attention heads significantly affects the model performance, as shown in Table 8 and Figs. 10 and 11. In general, increasing the number of heads improves the translation quality as it improves the model’s ability to capture various linguistic patterns and dependencies. For example, increasing the number of heads from 4 to 8 in a shallow model (N = 1) increases the BLEU score from 35.96 to 38.52. However, this positive trend does not hold for deeper models. When N = 6, increasing the number of heads to eight results in a BLEU score of only 13.14, which is much lower than the scores obtained with fewer layers. For N = 2 and N = 4, increasing the number of attention heads leads to a noticeable decrease in the BLEU scores, as seen in Table 8.

Figure 10 BLEU values obtained during the training process with layers (A) 1, (B) 2, and (C) 3 (without beam search).

Figure 11 BLEU values obtained during the training process with layers (A) 4, (B) 5, and (C) 6 (without beam search).

In particular, the BLEU score decreases from 47.34 (Head = 4) to 20.57 (Head = 8) for N = 2, while it decreases from 36.33 (Head = 4) to 27.55 (Head = 8) for N = 4. This decrease can be attributed to the increased model complexity brought by a larger number of heads, which can lead to optimization difficulties and difficulty in effectively learning the linguistic patterns of the training data. In addition, excessive attention heads can lead to overfitting, where the model cannot generalize well to unseen examples while over-focusing on certain features in the training data. This problem may be more pronounced in morphologically rich languages such as Turkish, where the diversity of word forms and sentence structures requires balanced attention mechanisms rather than overly complex configurations. This suggests that excessive attention heads may lead to overfitting or optimization difficulties in highly complex models. Dropout was applied to prevent overfitting and to try to make the model learn in a more generalizable manner. Layer normalization was also used to reduce optimization difficulties due to layer depth. Figures 6 and 7 further illustrate this issue, as BLEU scores during training without beam search show slower improvements for models with higher header counts and deeper layers and greater instability for shallower layer structures and lower headers. These results highlight the importance of balancing the number of attention headers with the model depth, especially for morphologically rich and complex languages like Turkish. Table 8 presents the BLEU scores obtained using Beam Search, while Figs. 10 and 11 shows the curves of the BLEU scores calculated at each period during training without Beam Search.

Figure 12 compares the performance of all Seq2Seq and Transformer models on BLEU under different configurations. In Transformer models, the effect of the number of layers and headers is evident; increasing the number of headers (Head = 8) generally increases the performance, while performance decreases in deep models (Layer 6). This is attributed to the fact that increasing the number of layers makes optimization more difficult, and the model is exposed to the risk of overfitting due to more complex structures. The figure emphasizes the importance of the balance between model architecture and data processing methods for optimal performance.

Figure 12 Comparison of different automatic evaluation metrics across all configurations.

To further investigate the qualitative performance and specific error patterns of the evaluated models, Table 9 presents a selection of English input sentences, their corresponding Turkish reference translations, and the outputs generated by each NMT configuration. As observed in Table 9, different models show different degrees of success depending on the complexity and length of the input sentence. For example, shorter and more direct sentences like ‘Are you feeling better today?’ (Input Sentence 1) are generally translated correctly by most configurations, especially those using BPE tokenization and attention mechanisms. However, the differences become more pronounced for longer and more complex sentences like Input Sentence 4. The Transformer WT model and the Seq2Seq Bahdanau BPE model generally produce more fluent and semantically consistent translations for these challenging inputs compared to their unattended WT counterparts or models. Overall, BPE models generally produce more fluent and semantically consistent translations for these challenging inputs compared to WT models. In particular, the Seq2Seq Bahdanau BPE model shows a remarkable ability to translate the reference translation for Input Sentence 4 almost perfectly, highlighting its strength in handling longer dependencies when combined with efficient subword tokenization. In contrast, the Seq2Seq Unattended WT model often produces disfluent or meaningless outputs (e.g., its translation for Input Sentence 4). The hybrid attention mechanism, while competitive, sometimes introduces minor changes or granularity compared to standard Bahdanau attention, as seen in the translation of Input Sentence 3 by the Seq2Seq Hybrid BPE. These qualitative examples highlight the importance of both the model architecture and the tokenization strategy. Although BPE generally leads to a more robust performance, it yielded inferior results when examined over the Transformer model. This is mainly due to the fact that morphologically rich languages such as Turkish increase the complexity in this model. In addition, attention mechanisms have been found to be crucial for maintaining consistency, especially in longer sentences.

Table 9 English–Turkish test translation samples.

Models	Translations	
Input sentence 1	Do you feel better today?	
Reference sentence 1	Bugün daha iyi hissediyor musun?	
Transformer BPE	Bugun daha iyi hissediyor musun?	
Seq2Seq Bahdanau BPE	Bugun daha iyi hissediyor musun?	
Seq2Seq hybrid BPE	Bugun daha iyi hissediyor musun?	
Transformer WT	Bugün daha iyi hissediyor musun	
Seq2Seq Bahdanau WT	Bugün daha iyi hissediyor musun	
Seq2Seq hybrid WT	Bugün daha iyi hissediyor musun	
Seq2Seq no attention WT	Bugün daha iyi hissediyor	
Input sentence 2	Some people say Boston is a dangerous city.	
Reference sentences 2	Bazı insanlar Boston’un tehlikeli bir şehir olduğunu söylüyorlar.	
Transformer BPE	Bazı insanlar boston a tehlikeli bir şehir olduğunu söyledi.	
Seq2Seq Bahdanau BPE	Bazı insanlar boston’ da tehlikeli bir sehir oldugunu soyledi.	
Seq2Seq hybrid BPE	Bazı insanlar boston’ da tehlikeli bir sehir oldugunu soyledi.	
Transformer WT	Bazı insanlar boston’ un tehlikeli bir sehir oldugunu söylüyor.	
Seq2Seq Bahdanau WT	Bazı insanlar boston a kadar tehlikeli	
Seq2Seq hybrid WT	Bazı insanlar boston a tehlikeli bir şehir olduğunu söylüyor	
Seq2Seq no attention WT	Bazı insanlar boston da tehlikeli bir şehir bir kişi var	
Input sentence 3	The doctor advised him to do more exercise.	
Reference sentence 3	Doktor ona daha fazla egzersiz yapmasını tavsiye etti.	
Transformer BPE	Doktor daha fazla egzersiz yapmasını tavsiye yaptı.	
Seq2Seq Bahdanau BPE	Doktor ona daha fazla egzersiz yapmasını tavsiye etti.	
Seq2Seq Hybrid BPE	Doktor ona daha çok çok egzersiz tavsiye etti.	
Transformer WT	Doktor ona daha fazla egzersiz yapmasını tavsiye etti	
Seq2Seq Bahdanau WT	Doktor ona daha fazla egzersiz yapmasını önerdi	
Seq2Seq hybrid WT	Doktor ona daha fazla egzersiz egzersiz yapmasını önerdi	
Seq2Seq no attention WT	Doktor çok egzersiz yapmasını etti	
Input sentence 4	A mistake young people often make is to start learning too many languages at the same time, as they underestimate the difficulties and overestimate their own ability to learn them.	
Reference sentence 4	Gençlerin sık yaptığı bir hata da; zorluklarını hafife alıp, kendi yeteneklerini de gözlerinde büyütürek aynı anda birçok dili birden öğrenmeye başlamaları.	
Transformer BPE	Genç insan oturup genellikle dil öğrenmek küçük bir tür kişiyi öğreniyor genç insan kadar çok zaman harcamaya.	
Seq2Seq Bahdanau BPE	Gençlerin yaptığı bir hata da; zorlukları hafife alıp, kendi yeteneklerini de gözlerinde büyütürek aynı anda birçok dili birden öğrenmeye başlamaları.	
Seq2Seq Hybrid BPE	Bir hata genc insanlar bir insan insanlar sık sık ogrenmek ve aynı zamanda ogrenmek cok sayıda sey calıstır maya basladı.	
Transformer WT	Gençlerin sık yaptığı bir hata da zorluklarını hafife alıp kendi yeteneklerini de gözlerinde büyütürek aynı anda birçok dili birden öğrenmeye başlamaları	
Seq2Seq Bahdanau WT	Gençlerin sık yaptığı bir hata zorluklarını hafife alıp kendi yeteneklerini gözlerinde büyütürek aynı anda dili birden öğrenmeye başlamaları	
Seq2Seq hybrid WT	Gençlerin sık yaptığı bir hata zorluklarını hafife alıp kendi yeteneklerini gözlerinde büyütürek aynı anda birçok dili birden öğrenmeye başlamaları	
Seq2Seq no attention WT	çok sayıda insan para almak için için onlar onun aynı dili yapmak ve o bilim adamı olduğu için hayal kırıklığına harcamak istiyor gibi ses izliyor	

Statistical significance tests. To further validate the observed differences in translation quality among our top-performing models, pairwise statistical significance tests were conducted on their BLEU scores. The methodology involved generating complete test set translations (hypotheses) for each of these models, saving them into separate text files alongside a single reference translation file (test.ref.tr), and then utilizing the sacrebleu library’s paired bootstrap resampling test (1,000 samples) to compare the BLEU scores of each model pair. This test computes a p-value, where a value below the 0.05 significance level indicates that the observed difference in scores is statistically significant and not due to chance. The comparison between Transformer WT and Seq2Seq Bahdanau BPE yielded a p-value of 0.0301, indicating a statistically significant difference in their performance. Similarly, the difference between Transformer WT and Seq2Seq Hybrid BPE was found to be statistically significant (p-value = 0.0394). Finally, when comparing the two Seq2Seq variants, Seq2Seq Bahdanau BPE and Seq2Seq Hybrid BPE, the p-value was 0.0472, also confirming a significant performance difference. These results provide strong statistical evidence that the variations in BLEU scores among these top-performing configurations are not due to random chance, reinforcing our conclusions about their relative effectiveness for the English-Turkish translation task.

Evaluation of the error metrics. In Fig. 13, the graph shows the performances of the Transformer model (Layer-3, Head = 8) and the Seq2Seq model (Bahdanau attention, DATABPE) regarding morphological, lexical, and word order errors. The transformer model provides superiority with lower rates in all error types. In particular, the morphological error rate was measured as 0.73 for the Transformer and 1.17 for the Seq2Seq model. Similarly, the lexical error rate was quite low at 0.47 for the Transformer, while it was recorded as 0.60 for the Seq2Seq model. In word order errors, Transformer also achieved better results with an error rate of 0.78 compared to Seq2Seq’s rate of 0.87. These results show that the Transformer model better captures local and global contexts thanks to its multi-head attention mechanism and deep architecture. As an agglutinative language, Turkish increases morphological diversity and complexity in word structures. This makes modeling correct contexts at the word level more difficult.

Figure 13 Comparison of the average error scores of Transformer model with different layer and BPE tokenization.

Although the Seq2Seq model’s hybrid attention mechanism offers improvements, Transformer’s attention mechanism captures Turkish agglutinative structures and contexts more accurately. In particular, the significant difference in morphological error rates reveals that Transformer offers an architecture more suitable for Turkish agglutinative structures. As a result, Transformer’s architecture increases translation accuracy and reduces error rates by processing grammatical and contextual relations in Turkish more effectively.

Conclusion

As an agglutinative language, Turkish presents challenges for translation models due to its morphologically rich and flexible syntax. The diversity of surface forms resulting from extensive suffixation complicates both word representation and semantic modeling. Additionally, Turkish’s flexible sentence structure increases the difficulty of accurate word order prediction. In this study, multiple NMT architectures, tokenization methods, attention mechanisms, and model configurations were evaluated to improve English–Turkish translation quality. Notably, BPE enabled more effective subword-level representation of morphological structures and led to higher BLEU scores. The best results were obtained using the Transformer model with three layers and eight attention heads trained on BPE-tokenized data, yielding the highest BLEU, METEOR, and TER scores among all configurations. attention mechanisms also contributed by enabling the models to capture both local and global contextual dependencies. Overall, the findings demonstrate that carefully optimized architectural and tokenization strategies can meaningfully enhance translation performance in morphologically complex and low-resource language settings. The findings of this study are particularly relevant for researchers working on neural translation for low-resource and morphologically complex languages, as well as practitioners aiming to adapt NMT models across linguistically diverse domains.

Limitations of the study

Despite its promising outcomes, this study has several limitations. First, although a relatively large English–Turkish parallel corpus was used, the data is domain-general and may not reflect translation behavior in specialized domains such as medical, legal, or conversational discourse. Domain-specific evaluations are essential to understand the strengths and limitations of the model in context-sensitive applications. Second, while the proposed methods were shown to be effective for Turkish–English translation, their applicability to other low-resource or morphologically complex language pairs has yet to be tested. Given that agglutinative and fusional languages present different challenges, the current models may require further adaptation. Future studies could extend the experimental framework to typologically diverse languages, such as Finnish, Hungarian, or Swahili, to examine generalizability. Third, an important direction for future research is the application of transfer learning strategies, where knowledge gained from high-resource or morphologically rich language pairs (e.g., English–Turkish) can be adapted to low-resource and typologically similar languages (e.g., Uyghur, Kazakh, or Azerbaijani). This would allow leveraging shared linguistic patterns such as affixation, vowel harmony, and word order. The use of pre-trained multilingual models (e.g., mBART, mT5) fine-tuned on morphologically aligned languages holds great promise in boosting translation performance under low-data regimes. Exploring these approaches can significantly enhance the scalability of NMT systems across lesser-studied languages. Finally, improving English–Turkish NMT systems can have important real-world implications. High-quality bilingual translation tools could serve as critical infrastructure for bilingual education, especially in regions where English is taught as a second language. Furthermore, they can enhance digital accessibility by providing more accurate translation of online resources, services, and public information for Turkish-speaking populations. In cross-border communication contexts, such tools could support international collaboration, migration services, and governmental or humanitarian communication efforts, thereby facilitating information exchange across communities with linguistic barriers.

Supplemental Information

Supplemental Information 1 Neural Machine Translation Codes.

Supplemental Information 2 Dataset.

In this study, AI tools (i.e., DeepL and GPT) were used for sentence corrections and translations.

Additional Information and Declarations

Competing Interests

The authors declare that they have no competing interests.

Author Contributions

Mehmet Acı conceived and designed the experiments, analyzed the data, prepared figures and/or tables, authored or reviewed drafts of the article, and approved the final draft.

Nisa Vuran Sarı conceived and designed the experiments, performed the experiments, analyzed the data, performed the computation work, prepared figures and/or tables, authored or reviewed drafts of the article, and approved the final draft.

Çiğdem İnan Acı conceived and designed the experiments, analyzed the data, authored or reviewed drafts of the article, and approved the final draft.

Data Availability

The following information was supplied regarding data availability:

The codes and data are available in the Supplemental Files.

The Turkish-English translation dataset used in this study is available at Kaggle: https://www.kaggle.com/datasets/seymasa/turkish-to-english-translation-dataset.

The majority of the sentence pairs in the dataset appear to be compatible with or derived directly from contributions from the Tatoeba Project (https://tatoeba.org/en/sentences/search?from=tur&query=&to=eng), a collaborative online database providing example sentences and translations for language learners and linguists.

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
