# Peer review of "Morphological and structural complexity analysis of low-resource English-Turkish language pair using neural machine translation models"

_PeerJ Computer Science, doi:10.7717/peerj-cs.3072_

## Round 0.1 · original submission · Major Revisions

Dear Authors,
Your paper has been revised. It needs major revisions before being accepted for publication in the PEERJ Computer Science. More precisely:

1) According to the results presented in Table 5, the transformer model with word-based tokenization performs better than the alternatives. However, the validity of the BPE-related results is uncertain, as the paper does not specify the vocabulary size used for BPE tokenization—a critical parameter for replicability and interpretation.

2) The revised version of the article should include
(a) a visual layout or workflow of the neural models used
(b) more information about the training details, such as batch size, learning rate, number of epochs, regularization methods applied, etc.
(c) More details about the tools, tokenizers, or segmentation strategies regarding morphological segmentation are needed.
(d) Statistical tests will be used to support the performance improvements of the proposed approach.

3) The use of Transformer and GRU-based Sequence to Sequence with Attention in the study about Neural Machine Translation Models is positive in terms of originality in terms of the study. However, in this section, especially about this problem solution, many different deep learning-based models can be preferred. You must provide more detail on the reasons for this choice.

4) The paper appears to use an older export from the Tatoeba challenge, yet the Tatoeba project is not referenced within the paper. Additionally, the dataset is not analyzed in depth, and critical details such as the size, number of tokens, number of unique words, and composition of the training and evaluation sets are not disclosed. All the above issues must be addressed.

Reviewer 1 ·

Basic reporting

The language and structure of the paper are generally understandable. However, there are several formal issues that need attention.

The text is not consistently organized into coherent paragraphs, which affects readability.
Acronyms are not always defined upon their first use.
The capitalization of proper nouns appears inconsistent and at times arbitrary.

The literature overview does not contain enough recent references to support the novelty of the contributions.

Experimental design

The experiments focus on evaluating the performance of English–Turkish machine translation systems based on encoder-decoder architectures using gated recurrent units (GRU) with attention mechanisms, as well as transformer-based models. The authors also compare word-based and byte-pair encoding (BPE)-based tokenization strategies.

According to the results presented in Table 5, the transformer model with word-based tokenization performs better than the alternatives. However, the validity of the BPE-related results is uncertain, as the paper does not specify the vocabulary size used for BPE tokenization—a critical parameter for replicability and interpretation.

Another set of experiments investigates the optimal number of layers and attention heads for the given training dataset. Table 6 indicates that three layers yield the best performance, while the number of attention heads appears to have a negligible impact.

A notable strength of the paper is the reported BLEU scores for Turkish, which are higher than those achieved by large-scale multilingual models such as NLLB 54B, despite the latter being trained on significantly larger datasets.

Nevertheless, the reported results raise some concerns.
1.) The translation task may be overly simplistic, as the dataset is derived from the Tatoeba challenge and consists predominantly of short, commonly used phrases.

2.) Furthermore, the data splitting procedure between training and evaluation sets is not clearly described, which compromises the reproducibility of the experiments.

3.) The authors do not provide a comparison with alternative approaches or baselines, and there are no experiments conducted for the reverse translation direction (Turkish–English).

4.) The vocabulary sizes for both source and target languages are not reported.

Validity of the findings

One notable aspect of the paper is the introduction of a hybrid attention mechanism that combines Bahdanau and Scaled Dot-Product attention. This idea may be innovative; however, both methods are well-established (originating in 2015), and more recent work has addressed their limitations. Unfortunately, the paper does not provide an adequate review of recent literature, nor does it include comparative experiments to convincingly support the originality or effectiveness of this hybrid approach.

A significant concern with the paper lies in the discrepancy between its stated contributions and the supporting evidence provided through experiments and literature review.

The authors claim that they provide " new perspectives to evaluate and improve the performance of neural machine translation". The paper does not sufficiently demonstrate how its evaluation methodology is novel. The experiments rely on an existing dataset and use standard evaluation metrics. The findings do not establish that the proposed methods outperform or are comparable to state-of-the-art alternatives.

Another serious issue concerns the dataset. The paper appears to use an older export from the Tatoeba challenge, yet the Tatoeba project is not referenced. Additionally, the dataset is not analyzed in depth, and critical details such as the size, number of tokens, number of unique words, and composition of the training and evaluation sets are not disclosed.

Additional comments

A commendable aspect of the paper is its focus on Turkish, a relatively low-resource language. Nonetheless, the major issues outlined above present substantial obstacles to the paper’s acceptance/

There are also several minor issues:

The abstract includes vague or underdeveloped claims:
“Turkish poses unique challenges for translation models” – The statement is not substantiated. Which linguistic features of Turkish contribute to these unique challenges?
“Two effective NMT architectures: Transformer and Sequence-to-Sequence (Seq2Seq) with Attention” – This needs clarification, as Transformer models are a subset of sequence-to-sequence architectures.
The distinction between automatic and statistical metrics is unclear. The paper should specify whether statistical metrics can be non-automatic or whether all automatic metrics are statistical.

The text formatting is inconsistent:
Paragraphs are poorly structured.
There are spelling mistakes.
Capitalization of terms is inconsistent.
Acronyms (e.g., TGRU, GRU, ChrF) are not explained upon first use.

Figure 1 is not meaningful as presented. The number of resulting tokens depends on the BPE dictionary size, which is an arbitrary parameter. A smaller BPE vocabulary leads to more tokens, and as the vocabulary size increases, the number of tokens approaches the original word count. The same critique applies to Table 2.

Reviewer 2 ·

Basic reporting

The article is well written.

Experimental design

The experiment design is adequate.

Validity of the findings

The findings in this work are scientifically validated.

Additional comments

This work is more like a technical report than an original research article. Specifically, the authors finetuned the famous deep learning architectures: Transformer and Seq2Seq models with Attention for the low-resource English-Turkish language pair. Moreover, I still find this work interesting, since it contributes to the literature of machine translation for understudied language pairs.

·

Basic reporting

Professional and clear English is used throughout the paper.
Figures and tables are appropriate, but additional visual aids could improve clarity.
Structure generally follows the PeerJ format.

Experimental design

The design is valid and in line with scientific norms, though more granularity is needed on architectural choices and hyperparameters.
The chosen metrics are appropriate, but more interpretation and statistical support are recommended.

Validity of the findings

Conclusions are grounded in the results, though slightly overstated in places.
Interpretation is generally sound but needs deeper error analysis and limitations discussion.

Additional comments

This article addresses a persistent challenge in neural machine translation—accurately modeling morphologically rich, low-resource language pairs such as English-Turkish. This study explores the structural and linguistic complexities involved and proposes techniques that potentially bridge performance gaps in these settings. Thank you for such a nice article.

Refer to peer review comments.

Abstract:
1. The abstract does not clearly communicate what is novel about this study. Please specify the unique methodological or empirical contribution.
2. Can you add concrete metrics or statistical outcomes? Instead of broad claims like "shows significant improvement”.
3. Can you briefly mention why studying English-Turkish is significant for NMT research, especially considering its morphological richness and low-resource nature?

Introduction:
1. Can you clearly establish research gaps? The current introduction lacks a focused articulation of what precise limitations in previous work that this study addresses.
2. Can you include a paragraph near the end of the Introduction explicitly outlining the research question or hypothesis?
3. Who is the target audience for this research: researchers in NLP, low-resource languages, or broader AI/ML?

Methodology / Experimental Setup:
1. Can you include a visual layout or workflow of the neural models used?
2. Can you add more information on training details, ex. Batch size, learning rate, number of epochs, and regularization methods applied, etc.
3. If you are using morphological segmentation, can you include more details about the tools, tokenizer, or segmentation strategies?

Results and Analysis:
1. Can you conduct statistical significance tests (ex., t-test, ANOVA) to support the performance improvements?
2. Can you provide qualitative examples where the model succeeded or failed for error analysis?
3. You may need to improve the results visualization, ex. Consider adding visual aids such as charts/graphs, bar plots, or confusion matrices for better clarity.

Conclusion:
1. The current conclusion overstates the study's implications without adequate reasoning/analysis.
2. Can you include any limitations, ex. Methodological limitations such as overfitting risks, domain-specific data, or limitations in generalizing to other language pairs.
3. Can you add specific directions for future research, such as transfer learning from morphologically rich to poor-resource pairs, multilingual training, or adaptation to unseen domains?
4. Can you discuss how better English-Turkish translation tools could impact bilingual education, cross-border information exchange, or accessibility?

Reviewer 4 ·

Basic reporting

All comments are presented in detail in the final section.

Experimental design

All comments are presented in detail in the final section.

Validity of the findings

All comments are presented in detail in the final section.

Additional comments

Review Report – PeerJ Computer Science
(Morphological and structural complexity analysis of low-resource English-Turkish language pair using neural machine translation models)

1. This study evaluates Transformer and Sequence-to-Sequence architectures with various tokenization and attention mechanisms for translating the morphologically rich Turkish language, revealing that a multi-layer, multi-head attention Transformer model with Byte Pair Encoding yields the most successful results.

2. In the introduction, the importance of the subject, what the Neural Machine Translation method is, and the models related to it, Sequence-to-Sequence and Transformer Attention structures, are basically mentioned. In addition, the main contributions of the study to the literature are clearly stated in detail and an explanatory manner.

3. The literature related to the study is discussed in the related studies section. However, when this section is examined in detail, it is observed that it needs to be detailed a little more. For this reason, to emphasize the main contributions in the previous section more and to summarize the current situation in the literature on Neural Machine Translation more clearly, it is recommended to add a more detailed literature table consisting of columns such as "method, model, originality, dataset, pluses, minuses" to this section.

4. The EN2TR dataset used in the study is both sufficient and very suitable for the scope of the study. The preprocessing of the dataset with two-step Tokenization processes, Subword-level Byte Pair Encoding, and Word-level Word Tokenization, before using it on the model has been very positive in terms of both the preferred preprocessing steps and increasing the quality of the dataset.

5. The use of Transformer and GRU-based Sequence to Sequence with Attention in the study in relation to Neural Machine Translation Models is positive in terms of originality in terms of the study. However, in this section, especially in relation to this problem solution, many different deep learning-based models can be preferred, Please provide a little more detail on the reasons for preferring them.

6. Both hyperparameters and their corresponding values are given in detail in relation to both Sequence to Sequence with Attention and the Transformer Model. The details in this section are very positive. However, here, the types and values of metrics such as learning rate, optimizer, and loss function should be interpreted more deeply with the reasons for which they were determined and/or different experiments were conducted. Since the results obtained may vary greatly depending on the types and values of these metrics, it is very important to explain this section.

7. When the Overall design is examined in detail within the scope of the study, it is observed that the study is expressed clearly and sufficiently. In terms of performance metrics, both the preferred metric types and the results obtained for each model corresponding to them are sufficient and clearly reveal the quality of the study.

8. The Conclusion section emphasizes the study with sufficient results. However, adding in-depth future works to this section will strengthen the place of the study in the literature and will greatly guide future studies and increase the potential for possible post-publication citations.

As a result, in this study, an analysis has been carried out using Neural Machine Translation, which has the potential to make a very important contribution to the literature in relation to English-Turkish Language. However, all the sections mentioned above should be taken completely.

---

## Round 0.2 · accepted · Accept

Dear Author,
Your paper has been revised. It has been accepted for publication in PeerJ Computer Science. Thank you for your fine contribution.

·

Basic reporting

This version looks better. Thanks for addressing the review comments.

Experimental design

This version looks better. Thanks for addressing the review comments.

Validity of the findings

This version looks better. Thanks for addressing the review comments.

Additional comments

This version looks better. Thanks for addressing the review comments.